# Recent Developments in Mpox Prevention and Treatment Options

**DOI:** 10.3390/vaccines11030500

**Published:** 2023-02-21

**Authors:** Shiza Malik, Tahir Ahmad, Omar Ahsan, Khalid Muhammad, Yasir Waheed

**Affiliations:** 1Bridging Health Foundation, Rawalpindi, Punjab 46000, Pakistan; 2Industrial Biotechnology, Atta ur Rahman School of Applied Biosciences, National University of Sciences and Technology, Islamabad 44000, Pakistan; 3Department of Medicine, Foundation University School of Health Sciences, Foundation University Islamabad, Islamabad 44000, Pakistan; 4Department of Biology, College of Science, UAE University, Al Ain 15551, United Arab Emirates; 5Office of Research, Innovation, and Commercialization (ORIC), Shaheed Zulfiqar Ali Bhutto Medical University, Islamabad 44000, Pakistan; 6Gilbert and Rose-Marie Chagoury School of Medicine, Lebanese American University, Byblos 1401, Lebanon

**Keywords:** MPXV, mpox, therapeutics, antiviral agents, vaccines, novel therapeutic, clinical management

## Abstract

Human mpox is an emerging epidemic in the world. The monkey pox virus (MPXV) belongs to the same family of zoonotic Orthopoxviridae as that of the smallpox virus and exhibits similar clinical symptomology. Information regarding its diagnostics, disease epidemiology, surveillance, preventive methods, and treatment strategies are being collated with time. The purpose of this review is to trace the recent events in the scientific platform that have defined new preventive and treatment strategies against mpox. A methodological approach has been used to gather data from the latest literature to comprehensively overview the emerging treatment options. The results portion will cover details regarding the prevention of mpox. It will also shed light on a brief description of contemporary vaccines and antiviral agents that have been evaluated for their treatment potential since the emergence of the mpox threat. These treatment options are setting the pace for controlling the widespread monkeypox infection. However, the limitations attached to these treatment strategies need to be tackled quickly to increase their efficacy so that they can be deployed on a large scale for the prevention of this epidemic becoming another pandemic in this decade.

## 1. Introduction

Despite so many advancements in the field of biotechnology, the world still has to face the burden of epidemics that slowly become pandemics. The world has yet not overcome the COVID-19 pandemic, and other viral epidemics are already here, such as mpox, Dengue, and Ebola among others [1]. The mpox virus is a zoonotic double-stranded DNA virus belonging to the family of Orthopoxviridae. Some other infectious viruses of the past such as poxvirus, variola, cowpox, and vaccinia virus also belong to the same family [2,3]. Most of these other viruses of this family have caused havoc in past decades due to their link with large-scale pandemics [4,5,6,7,8,9,10,11]. Therefore, scientists are worried about the evolution of an mpox endemic to the world [3]. So far, these are limited to African countries; however, some recent outbreaks have been reported in other continents as far as America, Asia, and Europe. The situation is thus alarming for scientific and healthcare networks around the world [12,13,14,15,16,17,18,19,20,21,22,23,24,25,26,27,28,29,30,31,32,33,34,35].

The main route of viral spread is through contact with infectious sores and scabs, sharing clothes and bedding, or through the bodily transfer of fluids [35]. The symptomology of mpox infection is similar to smallpox infection with a characteristic rash, fever, and flu-like symptoms [36,37]. According to WHO reports, approximately 68,000 disease incidences of mpox infection have been reported in endemic and outbreak regions. Moreover, 68 deaths have been recorded since the reporting of the first case of monkeypox infection [35,36,37,38]. At the time of writing this review, mpox has spread to 110 different countries of European, Asian, American, Middle Eastern, Australian, and South Asian origin [35,36]. The prevention, diagnosis, and treatment options for mpox are being developed from previous orthopox infection management. As it has a similar nature to smallpox and variola, the potential threat of the pandemic is the prime concern of the scientific community, and therefore vigorous effort is being made to deduce the effective therapeutic options and vaccination drives against monkeypox infection to avoid another biowarfare between humans and virus in the present decade [35,39,40,41,42,43,44,45,46,47].

At present, the mpox is still in limited endemic form with mild and self-limiting disease symptoms [48]. Therefore, in most places, supportive care is being provided to patients, and preventive techniques are being promoted to avoid further virus spread in communities [49]. However, the scientific community is still working continuously to compose antiviral therapies and useful vaccines ahead of time to tackle any future threats possibly arising from mpox [50,51,52,53,54,55,56,57]. As of now, two vaccines have gained FDA approval, namely JYNNEOS^TM^ and ACAM200^®^, which belong to live replication-incompetent and replication-competent vaccinia viruses, respectively [58,59]. Moreover, some antiviral agents such as tecovirimat, brincidofovir, and cidofovir are also being suggested for tackling mpox in severe diseased cases and immunocompromised patients [58,60]. The purpose of this article is to briefly evaluate the work carried out in the direction of prevention, treatment, and therapeutic compliance with mpox outbreaks and to discuss some of the latest technologies that are being given due consideration for future research orientation in the field of vaccinology and pharmaceuticals. Moreover, light will also be shed upon the need to carry out more research and collaborative work for preventing mpox outbreaks in non-endemic countries.

## 2. Methodology

A methodological approach has been used to gather the latest data regarding the different dimensions of mpox prevention and treatment strategies, which allowed the inclusion of data from diverse, recent, and the most-cited sources of research studies.

### 2.1. Data Extraction and Search Strategy

We researched electronic sources such as Google Scholar, Pub Med, NIH (National Library of Medicine), Scopus (Elsevier), and Web of Science. Moreover, the official websites of the WHO, CDC, UNAID, and FDA were also used to obtain the statistical results and latest updates regarding mpox treatment efforts. As the study mainly incorporates the data regarding therapeutics and vaccines against MPXV, the major research terms were “Monkeypox virus”, “mpox”, “mpox epidemic”, “therapeutics against mpox”, “antiviral agents”, “vaccines against mpox”, “vaccination strategies”, “therapies against mpox”, “novel therapeutic approaches”, and some other linked search terms.

### 2.2. Inclusion and Exclusion Criteria

After a thorough analysis of the dates, abstracts, titles, and journals of research publications, they were made part of this review. The process of information gathering was not limited to a few studies but rather collected from research compilations in the form of original research articles, reviews, short commentaries, case reports, and letters to the editors. Finally, the search strategy was limited to incorporating data from 2018 to 2022 to add only the most recent advances related to mpox management, especially during the latest endemic updates of 2022.

## 3. Results

Extensive data are already present regarding mpox etiology, epidemiology, latest updates on its outbreaks, infection cycle, host viral interaction, and possible viral targets for treatment purposes in recent publications [1,61,62,63,64]. Therefore, in this section we will only discuss the prevention strategies and treatment options, including vaccination and antiviral agents, that have come forward during outbreak months. The focus of the article remains, and readers obtain maximum information to understand therapeutic interventions against monkeypox infection.

### 3.1. Monkeypox: Course of the Epidemic and Possible Reasons for Recent Decline in Cases

MPXVwas first reported as an infection of zoonotic origin in the DRC back in 1970. It mostly remains endemic to 11 African countries, mainly the DRC, for these past 50 years [3,22,23]. Vulnerable cases, including children, pregnant women, and the elderly with a suppressed immune system, are more prone to develop severe conditions [30,31,32]. Owing to its likely similarity with smallpox infection, the smallpox vaccine can be utilized against monkeypox as directed by the WHO. A few vaccine candidates, including the smallpox vaccine, vaccine candidates, and antiviral agents reported in; tabular format in this study, are currently being employed and tested for the treatment of mpox in endemic regions [33,36].

The endemic nature of mpox has slowly taken the form of an epidemic due to reported cases in different far-off countries, such as the reported outbreaks in Nigeria in 2022 [24]. Similarly, some cases have been reported in the UK, back in April 2022 [17]. According to the WHO, approximately 110 countries have already reported confirmed cases of mpox; these may include the UK, Spain, Portugal, Canada, Germany, Belgium, Italy, France, the Netherlands, Sweden, the UAE, the Czech Republic, Brazil, America, and most reported in the DRC [4,5,8,21,32,37,38,39,40,41]. However, it should be noted that the cases are pertinent to change on an everyday basis, and cases may be differently reported by the WHO and CDC by the time of this publication. Until the date, 31 January 2023, ~ 21163 confirmed cases of mpox have been reported from 29 EU/EEA countries. In the latest reports presented by the WHO, the regions of America (88.2%) and of Africa (5.7%) have reported the highest number of mpox cases. However, there is also an observed decrease in overall mpox cases, by approximately 12.7%, compared to previous months [42,43,44,45].

The possible reason for the declining number of cases might be that the high-risk core groups of individuals have been undergoing vaccination. This scenario has generated a vaccination-elicited immunity in high-risk core groups [4,5,6,7,8,9,10]. Mpox cases are thus declining naturally. Moreover, the effective public health measure adoption in these endemic regions has also limited the spread of mpox infections [4,7]. An additional factor that should be noted is that mpox has neither a lifelong survival nature nor has purely the characteristic nature of sexually transmitted diseases such as HIV [4,12,13,14,15,16,17,18,19,20,21,22,23,24,25,26,27,28,29,30,31,32,33,34,35]. The disease mostly remains self-limiting and resolves within 2–4 weeks of symptoms, as the host slowly develops immunity against it. Thus, it is taken as an acute infection. However, it remains still unknown whether the first time mpox elicits durable and protective immunity to protect the host from re-infection or else they required proper medication and vaccination to prevent future infections. Moreover, the nature of sexual transmission is still needed to be confirmed, since the high-risk group seems to have adopted some behavioral modifications such as limiting sexual encounters, which is suggested as the major reason behind the reduced spread of mpox [60]. Owing to these factors, there has been a natural decrease in mpox cases worldwide, though the situation keeps on changing every day [37,38,39,40,41,42,43,44,45].

### 3.2. Preventive Measures against Monkeypox

Whenever an infectious outbreak hits the world, the first step is to determine a prevention protocol against it. Such is the case for the mpox resurging outbreak. For now, the approved vaccine is a smallpox vaccine that has been checked against mpox and has FDA approval status against smallpox but is still limited to a specific group of personnel. Thus, further clinical trials are needed for confirmed approval against mpox [62,65]. Other preventive measures proposed by healthcare authorities such as the WHO and CDC are outlined in the following section [34,36,47,66,67,68,69,70,71,72,73,74,75,76].

Avoidance of direct zoonotic contact with animals that may carry mpox, such as monkeys and squirrels, etc.Avoid contact with objects that may be in touch with the infected animals outside of the infected places.Avoid contact with sick individuals with suspected and confirmed mpox since the infection easily spreads through body lesions.Proper sanitization and hand washing after contact with infected objects, places, and animals and use of personal protective equipment when encountering infected individuals.Thorough washing and proper cooking of animal meat products.Adopting careful measures during physical interaction.Isolation of infected individuals to avoid infection spreading to other persons.Wearing medical masks and gloves in cases of confirmed mpox.Proper disinfection and cleaning of infected places and hospital floors.Increase public awareness regarding risks of infection, preventive measures, and possible treatment options.People with an increased risk of developing infection such as medical staff, laboratory workers, scientists, response teams, healthcare workers, and captive animals must be subjected to pre-exposure vaccination to avoid infection spread.Captive animals with the infection must be separated from other animals with proper quarantine care.

### 3.3. Pre-Exposure Prophylaxis

In addition to the above-mentioned points, healthcare authorities including the Advisory Committee and Immunization Practices (ACIP) have recommended the pre- and post-prophylaxis vaccination for a specific group of people [74]. For pre-exposure prophylaxis, vaccination with the FDA-approved vaccines (ACAM2000^®^ and JYNNEOS^TM^) is recommended for people working for healthcare authorities with occupations where direct exposure to orthopoxviruses is predicted [72,77]. These may include laboratory technicians and workers, clinical personnel involved in viral disease management, response teams against outbreaks, and vaccination and diagnostic teams, as well as scientists researching on clinical samples of mpox. Thus, proper vaccination protocols should be proposed for ensuring the safety for these personnel.

### 3.4. Post-Exposure Prophylaxis

PEP is recommended upon unprotected contact with the skin mucous membrane of an infected person or with their bodily fluids, saliva, lesions, oral cavity, clothing, bedding, etc. [72]. It may also be needed for people undergoing close space-sharing for long lengths (around 3 h or more) with the infected person, which may expose them to viruses through aerosol secretions and viral presence in air particles [78]. Additionally, post-exposure vaccination is only recommended by the FDA and CDC for high-degree exposures where there is a possible risk of contracting the virus but not a predictive confirmation as in the case of directly exposed persons [77,78]. Additionally, the lack of protective gloves and medical masks or contact-used material without pre- and post-exposure sanitization is a condition that sensitizes and necessitates vaccination. In cases of uncertain exposure or lower exposure rates, the recommended measures are to undergo diagnosis or monitoring before PEP [72,78,79]. Transmission takes place with prolonged interaction with an infected animal or symptomatic individuals. Thus, with informed guidance from the CDC, post-exposure vaccination should be conducted after approximately 4 days and within a period of 4–14 days to avert disease development [77,79]. If conducted later than the two week period, the disease onset cannot be prevented; however, the disease can only be reduced [79,80].

### 3.5. Therapeutics and Vaccines

Enormous therapeutics have been proposed, with some having promising results against orthopoxvirus family members. These compounds have proven antiviral effects on smallpox treatment, but there are no confirmed results for mpox in human beings [74]. However, as the first line of treatment, these antiviral agents, vaccines, and drugs are being utilized to avert the spread of mpox until a properly approved vaccine arrives in the healthcare market [80].

### 3.6. Vaccination Efforts against MPox

As described earlier, it is too soon to expect a vaccine against mpox since the outbreaks have been quite recently reported in different countries; before the current outbreak, the mpox was limited to only a few endemic regions of Africa [28]. However, now that the virus is surging in different countries, scientists have increased the pace of research to deliver an effective vaccine specific against mpox along with efficacious antiviral drugs [2]. At present, smallpox vaccines which exhibit up to 85% protection are being used for mpox, which is a good line to start [62,81]. The epidemiological data indicate that most of the mpox-infected cases were those who did not receive smallpox vaccination in childhood or had never been infected with poxviruses or those who were born after the smallpox pandemic and eradication period [82]. Currently, only two smallpox vaccines have been approved for mpox: a brief report of both vaccines is discussed ahead.

Upon CDC recommendation, these approved vaccines are used in the form of pre-exposure and post-exposure prophylaxis for some specific groups of people as discussed before. Apart from the approved vaccines, some other vaccination trials are also going on. One such experiment was on the Dryvax vaccine and the vaccinia virus vaccine (MVA), which were checked individually and in combination to determine the immune system response. The response came in terms of the initiation of cellular and humoral immune responses [60,83]. Moreover, some experiments on animal models compared the effect of vaccination on disease symptomology. Vaccinated animals were healthier and exhibited fewer or no symptoms while the non-vaccinated animals exhibited various illness symptoms associated with MPXV infection [60,84].

Similarly, some animal models were checked for protein-based vaccination, and they experienced mild to severe symptoms of disease yet still survived, unlike those animals that received DNA fragments and could survive [85]. Similarly, some studies concomitantly used both DNA and protein-based vaccines and received good results in the form of lower symptom rates and disease resolution within a few days. Immune system incitation in the form of antibody production against B-cell-conserved epitopes of mpox was also observed in these animals [86,87,88]. Moreover, experiments have also been conducted to check the impact of passive immunization by the transfer of vaccinia-neutralizing antibodies which demonstrated effective immune system responses in receiving animals [34]. All these studies need to be further checked, and confirmation on human models is the next step to officially obtain a licensed vaccine against mpox. At present, only the approved vaccination is being used for pre- and post-exposure prophylaxis in high-risk individuals. The risk attached to the immune escape, viral mutation, and other strains of orthopox viruses make it imperative to work continuously on alternative vaccines specified against mpox [65,77].

### 3.7. A Brief Account of Approved Vaccines

Two approved vaccines, JYNNEOS^TM^ and ACAM2000^®^, are currently being utilized for pre- and post-prophylaxis in specific patients. These vaccines are not readily available and are limited to some endemic regions and most developed countries for precautionary use against mpox-reported cases. Apart from these two vaccines, another vaccine, Aventis Pasteur Smallpox Vaccine (APSV) (a replication-competent vaccinia vaccine), is authorized for emergency purposes in case the other two vaccines are not available or contradicted for application. Some characteristic features of both approved vaccines are shown in Table 1, and a brief account of other vaccine trials is summarized in Table 2 below.

### 3.8. Antiviral Therapeutics against MPox

The smallpox viruses caused havoc back in its pandemic times and is still considered a bioweapon for its associated lethality and infectivity. Owing to the similar nature of monkeypox viruses, scientists are putting a lot of effort to propose rigorous drugs against mpox. The suggested drugs are undergoing approval stages, and some are being used for the supportive care of patients [65]. Problems associated with vaccination restriction compel the healthcare authorities to heavily rely on these supportive drugs and antiviral agents in case of treatment urgency. Moreover, the disease cases which already developed symptomology of mpox vaccine remain ineffective for reducing the symptoms, and thus drugs come into play. The three most common drugs that are being utilized against mpox are named Cidofovir, Brincidofovir, and Tecovirimat. A brief account of these drugs is discussed below.

### 3.9. Cidofovir (Vistide^®^) against MPox

This is a broad-spectrum antiviral agent which acts to terminate DNA polymerase-based replication in the form of 5′-diphosphorylated metabolite and thus is effective against a wide range of DNA viruses. It has proven to be efficacious against viral infections such as HIV, vaccinia, mpox, smallpox, etc. [98]. The main routes of administration include topical and intravenous administration. The best feature of this antiviral agent is that it decreases the symptomology to avert lesions formation and reduces mortality rates. It is used as a second-line therapy for severe vaccinia [82,99]. Specifically, in the case of mpox, the trials have demonstrated reversed mpox by inhibiting mpox replication (proven in vitro and in vivo experiments) [65]. However, it has been reported to cause nephrotoxicity during intravenous administration and thus requires probenecid and hydration and proper dose adjustment with renal functional considerations to deal with nephrotoxicity. Moreover, more clinical experimentation is required on mpox cases in humans [48].

### 3.10. Brincidofovir (Formerly CMX001) against MPox

Brincidofovir functions by phosphorylation to its active metabolic form “cidofovir diphosphate”. It selectively inhibits activity against orthopox DNA polymerase and thus has proven effective against DNA viruses. It is also known as a lipid conjugate of cidofovir and is considered to be a cidofovir diphosphate prodrug. Lipid acylic nucleoside phosphonate works as a phospholipid in the body and lessens lesion formation with no proven impact with coupled vaccination [81,82,83]. In some cases, reports demonstrated complete recovery of patients with minimal side effects. Thus, it received approval against cytomegalovirus retinitis in HIV patients and received approval for smallpox and related infections in 2021. It is preferred for rapid and widespread administration in cases of emergency outbreaks. Unlike cidofovir, it has good oral bioavailability and no reported nephrotoxicity; therefore, it is approved for oral administration [82,83,99]. However, the liver enzymatic profile must be carefully regulated, and functional tests must be conducted for the application. Additionally, due to hyperactive drug accumulation, it not recommended for immunocompromised patients, pregnant women, and newborns [43]. Recently animal experimentation against mpox reported effective treatment, but it requires more clinical experiments for proper approval and licensure [79,84].

### 3.11. Tecovirimat (TPOXX)-(ST-246) against MPox

This was licensed in 2018 for smallpox and in 2022 for mpox [64]. It functions to inhibit viral egress by targeting a unique gene that produces the m37 (F13L) envelope protein required for viral maturation and release [85]. It has reported high efficacy against orthopoxvirsues by reducing viral replication pace and viral load. Moreover, it causes delayed viral infection onset and decreases lesion formation and thus causes reduced infection symptomology and mortality rates [86]. Repeated experiments have well established its safety, tolerability, and pharmacokinetic profiles. It showed concomitant immunological effects in coupled vaccination experiments. However, the issue of drug resistance development is there, and no teratogenicity has been predicted in pregnant women [87]. Specifically for mpox, it has been checked in humans but lacks randomized phase 3 trials and thus needs further experimentation [88].

### 3.12. Vaccinia Immune Globulin (VIG) against MPox

This was licensed in 2018 against vaccinia virus infections. It acts as an alternative to antiviral agents. Its derivatives, recombinant immunoglobulins (rVIG), are under investigation [34,49]. rVIGs work on a strategic passive immunotherapy approach. The candidate’s plasma-derived vaccinia immunoglobulins (VIG), VIGIV Cangene and VIGIV Dynport, are under investigation and are not licensed yet [65,88]. It has been used successfully in refractive cases. As a hyperimmune globulin, it functions to neutralize virus particles and reduces viremia and mortality rates by up to 30–40% [57,77,89]. Antibodies are collected from the plasma of smallpox-immune individuals to create passive immunity. It has been redistricted for application in immunodeficient individuals as it contains amounts of maltose that possibly affect glycemic conditions and insulin levels, may interfere with serological testing, needs caution for renal insufficient profiles, and increases revaccination needs [43,79,84]. There are a lack of human testing data against mpox and thus needs further investigation. Apart from these three antiviral agents, some other important drug candidates under trial against mpox are summarized in Table 3. Note that a detailed explanation of these trials can be traced in the referenced publications since a detailed analysis of these studies is beyond scope of this review article.

### 3.13. Future Directions for Antiviral Therapies against Mpox

As progress in the field of biotechnology increases, more effort is being put forward to propose novel therapeutic approaches against emerging infectious diseases. Knowing the scope of current targeted therapy approaches such as those based on microRNAs and silencing RNAs, there is work happening to find accurate targeted molecules against mpox [3,74]. Similarly, biomarker-based therapies are also in research annals as biomarkers offer an effective targeting and flexible drug design approach against different kind of diseases [89]. For viral diseases, the biobased approaches allow effective screening, diagnosis, prognosis, and mitigative vaccination and adaptive treatment measures [80,98]. Moreover, by integrating the in silico and bioinformatic statistical and molecular models with biomarker-based theory, more sophisticated therapies could be formulated with fewer side effects, better delivery, reduced resistance potential, and improved pharmacokinetic properties [2,48]. The next step in the development of better antivirals against infectious diseases such as mpox includes the discovery of better cellular targets, innovative drug-targeting strategies, and improved drug delivery mechanisms [95,96,97,98]. Important drug targets and biobased markers tested in different studies are part of Table 3. Most of these biomarkers have been proposed while keeping in consideration the genomic similarities between smallpox and mpox, which sets a trace route for future therapies [38,48,65,80,82,95,96,99].

Another important development that is predictive to develop better treatment options for mpox in the future includes nanotechnology-based therapies [90,97,98]. Specifically, silver nanoparticles have been investigated for their proven antimicrobial properties. AGNPs are being utilized to decrease the infectivity of mpox in different studies [96]. Nanotechnology-based therapies offer novel, inexpensive, and broad-spectrum treatment options against different diseases. The basic principle is to alter the properties of present and newer antivirals at the nanoscale to improve their physiochemical features and linked pharmaceutical properties [74,96]. Another approach is the conjugation of nanoparticles with the approved drugs to improve the effectiveness, targeted delivery, and improved drug delivery to the body. Specific studies of AGNPs against mpox have exhibited their dose-dependent inhibitory effect. However, more studies are needed to prove the research implication in clinical models [3,48,65,95,99].

## 4. Conclusions

Complex and exaggerating burdens have been imposed on the healthcare system owing to the COVID-19 pandemic and converging outbreaks of various other viral infections at the same time including Ebola, Dengue, and mpox (the focus of the present study). Under these pressurized conditions, the scientific community is rigorously working to produce an effective therapy against mpox and other infectious diseases. Though there is effective work being conducted in the field of vaccination and drug design, the current developments are still considered theoretical because most of the drugs are for diseases of smallpox origin. Therefore, more effort should be kept on modern therapeutic options to specify them against mpox. There is a need to coordinate their efforts with an integrated approach with the medical industry for clinical experimentation. The recommendations from healthcare authorities have been properly outlined, and treatment recommendations and vaccination protocols have also been formulated against specific groups of the workforce and susceptible persons. Thus, a coordinated and participatory approach will be needed to educate the community and general public in awareness programs regarding mitigation, adaptation, and disease management before the epidemic of mpox becomes another pandemic of the century.

## Figures and Tables

**Table 1 vaccines-11-00500-t001:** A brief comparison of characteristics of approved mpox vaccines.

IMVAMUNE/JYNNEOS^TM^ Vaccine	ACAM2000^®^ Vaccine
Vaccines based on live attenuated and replication deficient mechanism of action [38]	Belongs to the category of live vaccinia virus [38]
No reaction at the inclusion site, no risk of inadvertent or autoinoculation reactions, and low risk of advertent transmission or vertical transmission due to replication incompetence [49]	Cutaneous reaction at the inoculation site. Risk of inadvertent and autoinoculation and progressive vaccinia and eczema development especially in immunocompromised. Additionally, the transmission risk for the replicative potential [49]
Replication incompetent strains of vaccinia Ankara-Bavarian Nordic (MVA-BN strain) are used [38,49,58,59,60,77]	The vaccinia virus strains are kept replication competent [77]
It is referred to as a “third-generation vaccine” [58,59,60]	It is referred to as a “second-generation vaccine” [77]
Can be used even in immunodeficient persons [58,59,60]	Reduced symptoms but possible side effects in immunocompromised people [89,90,91,92,93,94,95,96,97].
Not for the general population but for pre- and post-exposure prophylaxis.	Not available for the public even in endemic regions.
Storage Freeze-dried/subcutaneous	Storage Lyophilized/scarification
Authorized by EMA in 2013 and FDA in 2019 for the general population [83]	Received FDA licensure in 2007 for a specific group of populations [97]
Approved for smallpox with the name IMVANEX^®^, used for 18 years or older; 85% effective against mpox [81]	It replaced orthopoxvirus vaccine Dryvax [89]
It is produced by modified vaccinia Ankara-Bavarian Nordic (MVA-BN) strains that are grown in cell culture of primary chicken embryo fibroblast (CEF) cells.	It is manufactured by allowing NYCBH strain culturation in Vero cells.

**Table 2 vaccines-11-00500-t002:** Two other vaccines candidates in trials against mpox.

Vaccine	Replicating Potential	Storage/Route	Regulatory Status
Dryvax^®^	Yes	Freeze-dried/scarification	Discontinued manufacturing in the 1980s. Replaced with the new vaccine
LC16m8	Yes	Lyophilized, scarification	Licensed for use in Japan

**Table 3 vaccines-11-00500-t003:** Summary of some anti-viral agents against mpox.

Sr.no.	Drug Categories	Mechanism	Drug Formulations under Investigation	Importance Interventions	References
1	DNA polymerase inhibitors	Inhibit DNA replication	Nucleoside phosphonates Cidofovir and Brincidofovir HPMA and adenosine N1 oxide (ANO)	Ongoing trials, No effective tests from compounds such as acyclovir, brovavir, lobucavir, didanosine, ddC, or d4C, etc.	[82,98]
2	Inosine monophosphate (IMP) dehydrogenase inhibitors	Inhibition of viral replication	Ribavirin and Tiazofurin	Ongoing trials	[65,80,90]
3	S-Adenosylhomocysteine (SAH) hydrolase inhibitors	Inhibit viral replication	3-deazaneplanocin A (C3-NPC A) and carbocyclic-3-deazaadenosine (C-CA3-ADO)	High sensitivity to different viruses, potent antiviral effects exhibited in vivo tests, limited or no side effects	[98]
4	mRNA and protein synthesis inhibitors	Replication inhibition	33T57/Methisazone (Marboran^®^)	Not licensed yet, 30–40% efficacy, may promote side effects such as nausea and vomiting	[77,81]
5	Nucleoside analogues inhibitor	Inhibit the DNA replication process	Nioch-14	Easy to produce strong antiviral activity against many orthopoxviruses	[91]
6	Combination therapy	Combination therapy Live attenuated viruses +antiviral drug	ACAM2000 and Tecovirimat combination therapy	In vivo animal testing models (Cynomolgus macaques and Rhesus macaques)	[77,91]
7	combination therapy	Nucleotide analogs	Cidofovir and Elstree-RIVM	In vivo animal testing models (Cynomolgus macaques)	[58,77,83,91,98]
8	Antiviral agent	Nucleotide analog and a DNA polymerase inhibitor Modified cidofovir compound; inhibits DNA polymerase	CMX001	In vivo animal testing models (Rabbit)	[81]
9	combination therapy	vaccinia virus + F13L gene inhibition	ACAM2000 and Tecovirimat	In vivo animal testing (Cynomolgus macaques)	[59,81,90]
10	Antiviral agent	Inhibits release of intracellular virus	ST246	In vivo animal testing (*Cynomys ludovicianus*)	[59,81]
11	Immunomodulators	Immune system modulation	(A27L, VACV A33R, L1R, and B5R) Subunit vaccines and recombinant vaccine	In vivo animal testing	[34,81]
12	Antiviral agents	Inhibition of viral replication	RNA interference (siE8-d and siA6-a), Methisazone (Marboran), Hydroxyurea, DFBA	In vitro (cultured LLC-MK2 cells). Limited activities.	[49,65,88]
13	Bio-based markers	Targeted antiviral agents	Actin beta, Deoxythymidylate kinase, Annexin A1, Ubiquitin, Fi colin 2, Interferon-gamma, Interleukin 15, GTP cyclohydrolase I feedback regulator Membrane associated ring-CH-type finger 1, MNAT1 component of CDK Activating kinase, Makorin ring finger protein 3Specific peptidase 9 X-linked Retinoid X receptor alpha, IL-12A, Major histocompatibility complex, STAT3, Calpastatin cyclin dependent kinase 5, WASP actin nucleation promoting factor, SH3 domain binding protein 4, T cell receptor beta variable 20/OR9-2, TNF, Tyrosinase, Uroplakin 3B.	Ongoing trials	[2,3,34,35,38,48,64,65,85,92,93]

## Data Availability

Not applicable.

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
