# Peer review of "Recent Developments in Mpox Prevention and Treatment Options"

_vaccines, 2023, doi:10.3390/vaccines11030500_

Round 1

Reviewer 1 Report

This article aims to provide a review of the current status (as of end 2022, it seems) of vaccines and therapeutics relevant to the management of monkeypox virus infection, which is topically relevant and of interest to the global community. Unfortunately, there are substantial improvements needed to the methodology as well as the language before this article could serve its intended purpose.

1) The article includes information in the "results" on both vaccines and therapeutics but the search terms are only relevant to therapeutics (antivirals) without any key words relevant to vaccines, which seems to be a problematic omission. 

2) It is clear this is not meant to be a systematic review, but it is unclear once the searches were run, how articles / information was included or excluded. What was the article review and data extraction process?

3) The link between the methods and then the results presented should be strengthened. For example, the authors provide information on potential use of vaccines for both pre-exposure and post-exposure prophylaxis, but it is unclear how this information was sourced given it was not part of the overall search.

4) The "results" also include treatments in the table which are not relevant for mpox treatment, e.g. their mechanism of action is not relevant such as the case for reverse-transcriptase inhibitors, but then these classes of drugs are not explained in the text. If the authors want to include these for completeness purposes there should be a clearer relationship between the table and the "result" text.

5) the article focuses on the need for more clinical trials specifically on the use of these medical interventions specifically against monkeypox as much of their data historically was generated against other orthopoxviruses, namely variola virus (smallpox). Though this is sensible, there are indeed ongoing trials and might be good for the article to reference these as it seems in appropriate for the audience to perhaps come away from this article not realizing that efforts are indeed underway.

Finally, monkeypox virus causes the infection. mpox is now the disease that is caused by infection with monkeypox virus. This distinction should be accurately represented in the article.

Reviewer 2 Report

Review of the publication entitled:

,,Recent Developments in Mpox (Monkeypox) Prevention and Treatment Options’’.

The purpose of this review is to trace down the recent events in the scientific platform that has defined new preventive and treatment strategies against MPox virus infection. A methodological approach has been acquired to gather data from the latest literature to comprehen sively overview the emerging treatment options. The results portion will cover detail regarding the prevention of MPox viruses. It will also shed light on a brief description of contemporary vaccines and antiviral agents that have been evaluated for their treatment potential since the emergence of  the MPox virus threat. A methodological approach has been acquired to gather the latest data regarding the different dimensions of MPox prevention and treatment strategies. We researched electronic sources such as Google Scholar, Pub Med, NIH (National Library of Medicine), Scopus (Elsevier), and Web of Science.

Dear Editor,

 I am pleased to accept the article for review.

It requires major revisions:

1)      I recommend the authors follow the “PRISMA” checklist and provide the detailed search strategy and PRISMA flowchart and describe their approach to selecting the studies in this review, including the measures for quality assessment and inclusion and exclusion criteria in the Method section.

2)       In the result section, I’d like to see a summary of the PRISMA flowchart a table including the summary of the included articles such as the name of the first author, year of study, country of origin, sample and target group, number of participants, and main findings.

3)      The English should be checked and improved by a professional entity.

Reviewer 3 Report

The main virtue of the paper is the extensive list of references. There is too much emphasis on future vaccines, many of which are in early phases of development.

The authors fail to discuss the course of the epidemic, and the possible reasons for the recent decline in new cases.

Author Response

Reviewer 3

The main virtue of the paper is the extensive list of references. There is too much emphasis on future vaccines, many of which are in early phases of development.

Response: thank you for the valuable appreciation

The authors fail to discuss the course of the epidemic, and the possible reasons for the recent decline in new cases.

Response: both these aspects have been added under the heading “Monkeypox; course of the epidemic and possible reasons for recent decline in cases”

Round 2

Reviewer 1 Report

The updates provided by the authors have improved the clarity of the manuscript, but there still remain some minor issues.

- Throughout – please note, the name of the virus is still monkeypox virus, which can be abbreviated as MPXV. The name of the disease caused by infection with MPXV is now called mpox in English. Consider adjustments in the text as you see fit. More information can be found here: https://www.who.int/news/item/28-11-2022-who-recommends-new-name-for-monkeypox-disease

- Page 3, Line 113: “pregnant ladies” – please consider more modern terminology, e.g. pregnant women.

- Page 3, Line 128: “As of 17 January 2023, 18 Mpox cases have been reported from seven EU/EEA countries.” – Please consider rephrasing. When you say as of date, it would be expected that you note the cumulative cases, which I do not believe is 18 cases…The cumulative case numbers are likely more relevant than case numbers on any given day. Consider adjustment to how you are presenting the epidemiological situation.

- Page 3, Line 137-139: “An additional factor that should be notified is that Mpox has neither a lifelong survival nature nor is repeatedly acquired like other sexually transmitted diseases [4].” – Please consider rephrasing, this language is not clear. Are you pointing out the fact that this is an acute/resolving infection, not a chronic disease? Are you pointing out that infection elicits durable and protective immunity from re-infection? Also, not sure how the cited reference #4 highlights the points made.

- Page 3, Line 145: “Whenever an infection hits the world, the first step is to determine a prevention protocol against it.” This makes it seem that mpox has recently been discovered, which is not the case. Consider adjustment in how you are framing. This is the case in other sections as well.

- Page 4, Pre-exposure Prophylaxis: you reference ACIP and the FDA. Perhaps would be relevant to clarify these are agencies in the United States, not global authorities?

- Page 4, Post-exposure Prophylaxis – The flow of this paragraph could be improved. You jump between general statements on transmission risk and recommended actions. As well, some specific comments:

"Post-exposure vaccination is only recommended for high-degree exposures where there is a possible risk of contracting the virus but not a predictive confirmation as in the case of directly exposed persons [77], [78].” Who has made this recommendation? You seem to cite journal articles.

“Transmission takes place with prolonged interaction with an infected animal and symptomatic individuals.” Perhaps this should be “OR” not “AND”?

You note that post-exposure prophylaxis is recommended approximately 4 days post exposure and then that if done later than 4-14 days, it may not be as effective. This is confusing. If administered on day 7 post exposure, for example, would this be considered timely or too late? Perhaps adjust your language on this topic for more clarity.

Page 6, Antiviral therapeutics against MPox: consider referencing Table 3 at the end of this paragraph noting that it is a more comprehensive overview.

Would also strongly suggest the editors of the journal review the English language and use throughout. There are several awkward/confusing words and phrases that make comprehension a bit challenging.

Reviewer 3 Report

The article is now fit for publication.

Author Response

Thank You so much for giving valuable positive comments for the paper.